# Amyotrophic Lateral Sclerosis Mechanism: Insights from the *Caenorhabditis elegans* Models

**DOI:** 10.3390/cells13010099

**Published:** 2024-01-03

**Authors:** Lili Chen, Shumei Zhang, Sai Liu, Shangbang Gao

**Affiliations:** Key Laboratory of Molecular Biophysics of the Ministry of Education, College of Life Science and Technology, Huazhong University of Science and Technology, Wuhan 430074, China; lilichen93@163.com (L.C.); 13526489225@163.com (S.Z.); liusai1013@163.com (S.L.)

**Keywords:** ALS, *C. elegans* model, cellular mechanism, therapeutic application

## Abstract

Amyotrophic Lateral Sclerosis (ALS) is a debilitating neurodegenerative condition characterized by the progressive degeneration of motor neurons. Despite extensive research in various model animals, the cellular signal mechanisms of ALS remain elusive, impeding the development of efficacious treatments. Among these models, a well-characterized and diminutive organism, *Caenorhabditis elegans* (*C. elegans*), has emerged as a potent tool for investigating the molecular and cellular dimensions of ALS pathogenesis. This review summarizes the contributions of *C. elegans* models to our comprehension of ALS, emphasizing pivotal findings pertaining to genetics, protein aggregation, cellular pathways, and potential therapeutic strategies. We analyze both the merits and constraints of the *C. elegans* system in the realm of ALS research and point towards future investigations that could bridge the chasm between *C. elegans* foundational discoveries and clinical applications.

## 1. Brief Introduction of ALS

Motor Neuron Disease (MND) constitutes a group of disorders, including, but not limited to, Amyotrophic Lateral Sclerosis (ALS), Progressive Spinal Muscular Atrophy, Primary Lateral Sclerosis, and Progressive Bulbar Palsy. These disorders share a common feature: damage to upper and lower motor neurons, resulting in the loss of essential motor function [1,2]. Typically, individuals diagnosed with MND manifest symptoms such as muscle wasting and limb weakness. In addition to motor impairments, they may also encounter challenges related to language and swallowing [3]. As MND progresses, most patients succumb to complications such as pneumonia or respiratory failure [4]. This review specifically addresses the intricate mechanisms underlying ALS, recognized as the most prevalent adult-onset neurodegenerative form of MND. We predominantly summarize the molecular and cellular pathways from studies conducted with the small model animal *Caenorhabditis elegans* (*C. elegans*).

ALS, commonly known as Charcot’s disease or Lou Gehrig’s disease, is a motor neuron disease (MND) affecting both upper and lower motor neurons within the central nervous system, governing voluntary muscle movement. Clinically, ALS is characterized by muscle rigidity and the gradual weakening of limbs and bulbar muscles, leading to varying degrees of difficulty in speech, swallowing, and respiration [4,5]. It is noteworthy that functions such as bladder control, bowel movements, and eye movements typically remain unaffected until the advanced stages of the disease [6]. In addition to muscle dysfunction, 30–50% of ALS patients present with cognitive and other nervous system deficits. Common cognitive symptoms in ALS patients include challenges in social cognition, verbal memory, language, and executive function [7]. Approximately 15% of cases with observed cognitive impairment exhibit visible atrophy in the frontal and/or temporal lobes, resulting in behavioral changes or language impairment meeting the criteria for frontotemporal dementia (FTD) [8]. These observations indicate the complexity of ALS as a progressive disease involving functional deficits in multiple tissues.

The onset of ALS commonly occurs between the ages of 40 and 70, although cases exist among younger patients. Individuals diagnosed with ALS typically experience a rapid progression of the disease, with a life expectancy ranging from 2 to 5 years [9,10]. The estimated incidence rate of ALS exhibits variability, ranging from 0.3 to 2.5 cases per 100,000 individuals [11]. Despite being perceived as relatively rare in the public consciousness, ALS significantly influences the quality of life for affected patients.

## 2. ALS-Associated Proteins

Protein misfolding and aggregation are prevalent phenomena in ALS, affecting more than 90% of patients [12]. This section provides a concise overview of the key genetic factors, including SOD1, TDP-43, FUS, C9ORF72, OPTN, and others, that underlie the molecular basis of ALS. Subsequently, we present an introduction to the disease mechanisms associated with these genetic factors as elucidated through the utilization of the *C. elegans* models.

SOD-1 (Superoxide dismutase 1), a widely expressed cytosolic protein, plays a crucial role in converting toxic superoxide anions to hydrogen peroxide, serving as a cellular defense against oxidative stress [13,14]. Initially linked to a loss of dismutase activity, it is now recognized that mutations in SOD1 primarily induce toxicity through a gain-of-function mechanism, although the precise toxic mechanisms remain incompletely understood [15]. Mutations in the SOD1 gene lead to alterations in the enzyme’s folding and stability, ultimately resulting in aggregation within motor neurons and subsequent neuronal death [16].

TDP-43 (TAR-DNA binding protein 43), encoded by the TARDBP gene on human chromosome I, is primarily localized within neuronal nuclei [17]. Functioning as a versatile RNA-binding protein, TDP-43 is intricately involved in RNA processing, encompassing the regulation of alternative mRNA splicing and mRNA stability [18]. Notably, TDP-43 stands as a pivotal constituent of ubiquitin-positive aggregates found within the motor neurons of ALS patients. These aggregates consist of C-terminally truncated and hyperphosphorylated TDP-43 [17]. Over 40 ALS-associated mutations have been identified, with G298S, A315T, M337V, G348C, and A382T being the most frequently observed mutations [19].

FUS (Fused in sarcoma), a DNA and RNA-binding protein, plays a significant role in regulating transcription and mRNA processing in neurons. Mutations in FUS lead to the formation of aggregated proteins, resulting in motor impairment and synaptic function alterations through overexpression [20,21]. At the molecular level, FUS mutations cause dysregulation in various RNA processes, including splicing, transcription, and stabilization, ultimately culminating in neuronal dysfunction [22,23].

C9ORF72 is the most prevalent genetic cause of ALS. Mutated C9ORF72 genes exhibit distinct expansion of the hexanucleotide GGGGCC repeat in the first intron [24]. While the precise mechanism of this expansion remains elusive, one interpretation suggests that AUG-independent translation of GGGGCC may lead to the formation of dipeptide repeats [25]. 

OPTN (Optineurin), a hexameric protein weighing 64 kDa and composed of 577 amino acids (aa), interacts with numerous proteins involved in processes such as inflammation, vesicle-based protein trafficking, and signal transduction, including the nuclear factor kappa B (NF-κB) pathway [26,27]. Associated with neurodegenerative diseases like ALS, OPTN comprises several coiled-coil domains, a ubiquitin-binding domain (UBD), a leucine-zipper kinase, and an LC3-binding domain [28]. Specific mutations in OPTN, such as exon 5 deletion, Q398 nonsense, and E478G missense mutations, have been identified in ALS patients [29,30]. 

## 3. Molecular Mechanisms of ALS in *C. elegans* Models

*C. elegans* has emerged as a potent tool for investigating the mechanisms underlying neurodegenerative diseases [16,31,32,33]. This is primarily attributed to its facile genetic manipulation, rapid cultivation, and its utility as a whole-animal system amenable to various molecular and biochemical techniques [34]. Moreover, *C. elegans* genes exhibit functional conservation with numerous critical pathogenic genes associated with ALS in humans, thereby emphasizing the biological relevance of models established in this organism [35,36]. Indeed, *C. elegans* has gained widespread recognition as an animal model for scrutinizing the fundamental causal genes of ALS [37,38,39,40,41]. Approximately 42% of human disease-associated genes have identifiable nematode orthologs, making this worm a fitting model for exploring the molecular mechanisms and cellular processes driving disease onset and progression [34,42]. In Table 1 below, we present an overview of the nematode ALS models employed in various studies.

In the pursuit of understanding the pivotal genetic factors underlying various forms of ALS, the development of in vivo models has proven indispensable. Two overarching strategies have been employed for generating these models. The first method involves the overexpression of human wild-type or mutated ALS-associated proteins in model organisms, allowing researchers to scrutinize the functional and structural consequences of these proteins in specific tissues. This approach reveals the underlying pathological mechanisms of human ALS-associated molecules in animal models. The second approach revolves around the creation of loss-of-function or gain-of-function ALS-related mutants through the manipulation of homologous genes in model organisms. This assists in introducing the intrinsic functions of the key molecules involved in ALS and comparing the similarities in the pathogenic mechanisms of different species.

Several transgenic *C. elegans* strains expressing human SOD-1 variants have been developed to investigate the functions of mutant SOD-1 [32]. Initially, an ALS *C. elegans* model was established by employing the *hsp16.2* or *mec-3* promoter to express human wild-type SOD-1 and various familial ALS (FALS)-related mutants (A4V, G37R, and G93A) of SOD-1. However, these transgenic nematodes did not exhibit any discernible phenotypes [44]. Another transgenic *C. elegans* model was then developed to explore differences in the aggregation and toxicity tendencies of wild-type and mutant SOD-1. This was achieved by introducing YFP-tagged wild-type or mutant (G85R and G93A) SOD-1 proteins into the body wall muscle cells under the regulation of the *unc-54* promoter, resulting in the appearance of heterogeneous populations of aggregates associated with mild cellular dysfunction [47]. These transgenic worms exhibited distinctive features, allowing for a nuanced examination of the aggregation and toxicity patterns linked to both wild-type and mutant SOD-1. Transgenic *C. elegans* with pan-neuronal expression of SOD-1(G85R) under the control of the synaptobrevin (*snb-1*) gene promoter exhibited severe locomotor defects and presynaptic dysfunction, correlated with the insoluble aggregation of SOD-1 in neurons [51]. When mutant SOD-1(G93A) was expressed in GABAergic motor neurons, it led to age-dependent motor impairments, axon guidance failures, and significant SOD-1 accumulation [41]. The *C. elegans sod-1* gene shares functional similarities with its human counterpart. In SOD-1 loss-of-function mutants, the levels of superoxide anions were increased, resulting in a shorter lifespan and heightened susceptibility to certain environmental stresses [52]. Conversely, over-activation of *C. elegans sod-1*, as a by-product of the catalase reaction, elevated hydrogen peroxide levels and extended lifespan [39].

The *C. elegans* ortholog of TDP-43, denoted as TDP-1, shares molecular characteristics with its mammalian counterpart. TDP-1 exhibits a high affinity for binding to the canonical TDP-43 binding sequence [(UG)n] and is capable of substituting for human TDP-43 in in vivo splicing assays, suggesting the conservation of its fundamental molecular functions [53]. The initial *C. elegans* model for TDP-43-related ALS was established by inducing pan-neuronal expression of human TDP-43 under the control of the *snb-1* promoter. This resulted in pronounced phenotypes, including coordinated slow movement and defasciculation of the GABAergic motor neurons [53]. Recent advancements in transgenic *C. elegans* models have expanded to encompass pan-neuronal expression of both wild-type and mutant human TDP-43 variants, such as G290A, A315T, and M337V [54]. These investigations have revealed that wild-type TDP-43 induces moderate motor defects, while mutant TDP-43 variants precipitate severe motor dysfunction [54]. Intriguingly, analogous phenotypes to those observed upon wild-type TDP-43 overexpression were replicated when the C-terminal fragment of human TDP-43 was pan-neuronally expressed [50]. 

Numerous transgenic models of *C. elegans* have been generated to investigate the effects of mutated and overexpressed FUS genes [48,55,56]. *C. elegans* possesses an ortholog of FUS, known as FUST-1, sharing approximately 50% identity at the protein level. Previous studies reported that deletions in *fust-1* led to neuronal degeneration and paralysis, while overexpression did not exhibit any discernible effects [57]. Studies in *C. elegans* have unveiled structural and functional similarities between FUS and TDP-43, showing comparable outcomes when mutated ALS-associated variants are expressed [48]. Specific ALS-related mutations, R524S and P525L, in the FUS gene have been employed to establish transgenic *C. elegans* models, demonstrating impaired neuromuscular function and locomotion reminiscent of ALS characteristics [38]. Recent advancements in genetic manipulation have allowed the development of a single-copy FUS mutant transgenic strain of *C. elegans*, manifesting ALS-like phenotypes, including GABAergic neurodegeneration and progressive paralysis [58]. Interestingly, neurons with FUS-positive inclusions exhibited substantially reduced expression levels of dynactin 1, a retrograde motor protein, suggesting an association between nucleocytoplasmic transport perturbation and the formation of cytoplasmic FUS inclusions in sporadic ALS [59]. Furthermore, it has been proposed that reducing dynactin-1 levels can disrupt autophagosome transport and induce motor neuron degeneration. Building on this insight, Ikenaka and their team have developed a novel *C. elegans* transgenic model and found that *dnc-1* (*C. elegans* dynactin 1) knockdown disrupts the transport of autophagosomes, inducing motor neuron degeneration [60]. Notably, this behavior-based model has been employed to identify and assess potential neuroprotective drugs against motor neuron diseases, opening up new avenues for drug discovery [61].

*Alfa-1*, an ortholog of the ALS/FTD-associated gene C9ORF72 in *C. elegans*, provides a valuable model to explore its association with ALS [40]. The development of a transgenic model inducing motor deficits in *C. elegans* began with a mutation in the *alfa-1* ortholog. This model unveiled a synergistic toxic effect when combined with a TDP-43 mutation [37]. *C. elegans* with loss-of-function mutations in the *alfa-1* gene exhibit age-dependent motility impairments, ultimately leading to paralysis and GABAergic stress-dependent neurodegeneration [31,37]. Notably, *alfa-1* mutants manifest endocytosis defects, partially rescued by the expression of the human wild-type C9ORF72 protein, highlighting a degree of functional conservation [62]. However, it is essential to note that transgenic nematode models predominantly investigate human C9ORF72 toxicity, as *alfa-1* lacks hexanucleotide repeat expansions [31]. Comparative studies using models with different repeat lengths have disclosed that transgenes with 29 GGGGCC repeats induce early-onset paralysis and increased lethality. These phenotypes were absent in wild-type animals and those expressing the empty vector, and were less severe in animals with 9 GGGGCC repeats [63,64].

Mutations in the OPTN gene encoding optineurin have been identified in ALS patients. Three distinct types of OPTN mutations, including a homozygous deletion of exon 5, a homozygous Q398X nonsense mutation, and a heterozygous E478G missense mutation within its ubiquitin-binding domain, are linked to ALS pathogenesis [30]. Notably, expression of nonsense (Q398X) and missense (E478G) OPTN mutations in patients promotes inflammation and induces neuronal cell death by activating NF-kB [65]. Moreover, the E478G mutation resulted in a puncta-like aggregated distribution, differing from the dispersed cytoplasmic distribution of wild-type OPTN or a primary open-angle glaucoma (POAG) mutation [29]. Hence, OPTN plays a significant role in the pathogenesis of ALS, and targeting NF-kB with inhibitors could potentially be a therapeutic approach for ALS treatment. It is interesting that the absence of transgenic models related to OPTN in *C. elegans* has been overlooked. This may be because no *C. elegans* homolog of OPTN has been identified until now.

We summarize the ALS pathogenic mechanisms in Table 2.

## 4. Pathogenic Mechanisms of ALS Implicated in *C. elegans*

(1)Innate immunity

Data from clinical studies show that multiple genetic mutations linked to ALS enhance neuroinflammation, which provides compelling evidence for immune dysregulation in the pathogenesis of ALS [30,79,80]. Although *C. elegans* lacks a classical inflammatory response or inflammatory cytokines analogous to mammals, it possesses an innate immune system. Notably, mutated ALS-associated proteins have been found to activate an innate immune response in *C. elegans* [81,82,83]. In *C. elegans* strains expressing mutant TDP-43 or FUS in their motor neurons, age-dependent motility defects culminate in paralysis and motor neuron degeneration at a rate significantly higher than that observed in wild-type TDP-43 or FUS control strains. By examining the expression of immune response proteins, including NLP-29 (an antimicrobial, neuropeptide-like protein expressed in hypodermal and intestinal tissue), it is evident that the expression of mutant TDP-43^A315T^ or FUS^S57∆^ protein triggers the upregulation of immune response genes, suggesting that innate immune response may contribute to motor neuron neurodegeneration. Furthermore, mutated ALS-associated proteins trigger an TIR-1/Sarm1 immune pathway innate immune response in *C. elegans* motor neurons [84,85]. Loss-of-function mutations in *tir-1*, associated downstream kinases, and the transcription factor *atf-7* collectively serve to suppress motor neuron degeneration, further supporting the notion that the innate immune system is involved in ALS models. 

Despite knowing the importance of the immune response in ALS, there are many details that have not yet been elucidated. For instance, mutated ALS-associated proteins in neurons may elicit an immune response as part of a host defense reaction against pathogens or aid tissue repair. The molecules necessary to induce the expression of NLP-29 also need to be further explored. Chikka et al. reported that activation of the mitochondrial p38MAPK/ATF-7 immune pathway in the intestine is neuroprotective and sufficient to prevent rotenone-induced neurodegeneration [86]. Due to mitochondrial dysfunction being a prevalent feature of many neurodegenerative diseases, including ALS [87], the mitochondria-regulated immune pathway may also be involved in *C. elegans* motor neuron degeneration. *C. elegans*’ innate immune response coordinates its activity with the insulin/IGF-1 pathway [88], suggesting that insulin-related immune pathways are also worth investigating. Nevertheless, these studies reveal that cell-based strategies that enhance anti-inflammatory reactivity and reverse immune dysregulation offer the potential to slow disease progression and improve the quality of life of patients with ALS.

(2)Autophagy

It is widely acknowledged that the dysregulation of autophagy in motor neurons is a pivotal event in ALS [89,90,91]. Particularly, intensified immunoreactivity in the cytoplasm of motor neurons for microtubule-associated protein 1 light chain 3 (LC3), which is a marker of autophagosome, is frequently observed in the spinal motor neurons of ALS patients [92,93]. Consistent with this observation, *C. elegans* with dynactin 1 knockdown (*dnc-1(RNAi) worms*) in ventral motor neurons under the control of the *Pacr-2* promoter exhibited notable motor impairments, coupled with axonal and neuronal degeneration. Notably, the autophagosomes were easily trapped where the axon was tight, curved, or at spheroids. The phenomenon was followed by the accumulation of autophagosomes distal to the trapped sites [60]. Given that autophagosomes serve as cargo for dynein/dynactin complexes and play a pivotal role in the turnover of various organelles and proteins, the accumulation of autophagosomes suggests a potential contribution of dysfunctional autophagy to motor neuron degeneration in ALS. Indeed, the introduction of pharmacological disruptions to autophagy, using 3-MA, resulted in locomotory defects and axonal degeneration mirroring those observed in *dnc-1(RNAi)* worms. This implies that a compromised autophagy system alone is adequate to induce motor neuronal degeneration [60].

The contribution of defective autophagy to neuronal dysfunction in ALS is well-documented by autophagy-related genes [30,94,95,96,97,98]. The selectivity of autophagy is mediated by autophagy receptors that recognize and deliver cargoes to autophagosomes for degradation. SQSTM1/p62 is an autophagy receptor that is commonly found in protein aggregates in ALS brains. Related reports showed that SQSTM1 promotes the clearance of stress granules, a hallmark of ALS, via selective autophagy [99,100]. The main autophagy process is proteotoxic stress, which activates serine/threonine kinase TBK1, promotes phosphorylation of autophagy receptor SQSTM1, and activates selective autophagy. In contrast, ALS-linked mutations of TBK1 or SQSTM1 reduce SQSTM1 phosphorylation and compromise ubiquitinated cargo binding and clearance (Figure 1). The accumulation of SQSTM1 implicates a disturbance of the selective autophagy pathway [101]. Corresponding with the accumulation of autophagosomes, SQSTM1/p62 has been observed to accumulate in the motor neurons of ALS patients [102]. This observation aligns with findings demonstrating elevated levels of LC3-positive autophagy vesicles in the motor neurons of ALS patients with FUS mutations [103]. Notably, in an ALS *C. elegans* model involving overexpressing human mutant FUS proteins, a gain of toxic function mechanism disrupts basal neuronal autophagy. There was an increased accumulation of SQST-1 that disrupts neuromuscular function in stress conditions, the *C. elegans* ortholog for SQSTM1/p62, in motor neurons [38], reinforcing the link between autophagy dysfunction and ALS pathology. Conversely, the loss of *sqst-1* suppresses both neuromuscular and stress-induced locomotion defects in FUS-associated ALS worms. It is worth mentioning that this suppression likely does not accompany a correction of neuronal autophagy defects [38], indicating that SQST-1 operates through an autophagy-independent pathway or alternative mechanisms to ameliorate ALS-related locomotion impairments [104,105]. The mutation of a single autophagy receptor can induce the decline of autophagy and lead to abnormal protein accumulation. But when autophagy receptors are passively increased, reducing autophagy levels may have a positive effect. Thus, the treatment of ALS requires multi-factorial and systematic consideration.

Furthermore, increased expression levels of autophagic genes by *daf-2(e1370)* have been shown to protect *C. elegans* motor neurons against the toxicity of human SOD-1(G93A) [41]. Metformin, the globally prescribed biguanide drug worldwide for the treatment of type II diabetes, alleviates motor dysfunction in human SOD-1(G93A)-associated ALS worms, partly through enhancement of autophagy [106]. Although not explicitly validated in *C. elegans* models, investigations in other model systems have demonstrated cross-regulation between TDP-43 pathology and autophagy [107,108]. These findings imply the existence of supplementary autophagy mechanisms in ALS [109]. In conclusion, enhancing autophagy emerges as a novel and significant therapeutic target for addressing motor neuron degeneration in ALS.

(3)Protein homeostasis

Protein homeostasis (proteostasis) is carefully maintained through a finely regulated and interconnected network of biological pathways, crucial for preventing the accumulation and aggregation of damaged or misfolded proteins [110]. Conversely, the breakdown of proteostasis has been implicated in the etiology of various neurodegenerative diseases, including ALS [111]. Zhang et al. conducted genetic analysis and expression profiling of loss-of-function *tdp-1* mutants, elucidating the role of *C. elegans* TDP-1 (nematode TDP-43 ortholog) in regulating protein homeostasis. In diverse proteotoxicity models, the loss of TDP-1 alleviated protein aggregation and neuronal dysfunction. Their findings suggest that TDP-1 loss may modify global RNA levels, consequently impacting protein homeostasis and prompting cellular adaptation to stress on protein quality control systems [70]. Transgenic *C. elegans* models that express human TDP-43 variants displayed severe locomotor defects associated with the aggregation of TDP-43 in neurons [50]. Notably, the neurotoxicity and protein aggregation of TDP-43 were influenced by environmental temperature, and heat shock transcriptional factor 1 (HSF-1) played a role through protein quality control [112], indicating that a deficiency in protein quality control serves as a risk factor for TDP-43-associated ALS [50]. 

Aging-related neurodegeneration associated with TDP-43 is further connected to protein misfolding. The well-explored regulatory mechanism governing longevity and proteostasis involves the modulation of the insulin/IGF-1 signaling pathway through phosphorylation [113,114]. In the nematode *C. elegans*, the downstream receptor of insulin molecule, *daf-2*, has demonstrated the capability to counteract the shortened lifespan resulting from FUS overexpression [50,115]. These findings illuminate the intricate network of cellular mechanisms, notably the insulin/IGF-1 pathway, designed to preserve protein homeostasis in the presence of environmentally induced damage or genetically encoded misfolded proteins.

(4)Energy metabolism

While the intricate mechanisms underlying the association between mutant TDP-43, FUS, and Amyotrophic Lateral Sclerosis (ALS) are intricate and multifaceted, an accumulating body of evidence supports the presence of dysregulated energy metabolism in both ALS patients and relevant models [116]. The AMP-activated protein kinase (AMPK) serves as a pivotal cellular energy sensor. Upon activation, AMPK restores energy homeostasis by facilitating catabolic pathways, thereby promoting ATP generation [117]. Notably, heightened AMPK activation has been documented in the motor neurons of ALS patients, displaying a notable correlation with the extent of cytoplasmic mislocalization of TDP-43 [118]. These observations establish a clear link between energy depletion in human motor neurons and the pathological presence of TDP-43 in ALS. In line with this correlation, reducing AMPK activity has been demonstrated to ameliorate disease outcomes both in vitro and in *C. elegans* models expressing mutant SOD1 or TDP-43 [45]. Although a definitive mechanistic link between AMPK-regulated energy metabolism and TDP-43 mislocalization remains elusive, a proposed hypothesis suggests that the aggregation of TDP-43 stems from AMPK-mediated inhibition of nucleocytoplasmic transport.

Mitochondrial dysfunction is a prevalent characteristic of ALS [119,120]. Mutant forms of TDP-43, SOD-1, and FUS proteins have been implicated in disrupting mitochondrial structure and function [121,122,123]. This dysfunction induces an energy imbalance within neurons, affecting energy production and utilization. Specifically, FUS mutations have been linked to disturbances in mitochondrial function, potentially impeding the neurons’ capacity to generate ATP [121]. Consequently, compromised energy metabolism may play a role in the vulnerability and degeneration of motor neurons in ALS. Enhancing mitochondrial biogenesis emerges as an appealing therapeutic strategy for ALS. It is important to note that, despite the absence of current evidence from the *C. elegans* model, gaining further insights into these altered physiological processes in neurons—particularly by expressing mutant TDP-43 or FUS in *C. elegans*—becomes crucial for a more comprehensive understanding of ALS pathogenesis, specifically pertaining to energy metabolism.

In summary, mitochondria play a key role in ATP supply to cells via oxidative phosphorylation. Decreased ATP levels emerge as a common feature in ALS. It is conceivable that, in line with the high energy demands of neurons, gradual depletion of ATP, due to reduced respiration, may trigger neuronal degeneration [87,116]. In addition, the mitochondrial REDOX reaction is associated with the production of SOD. A lower concentration of ROS is essential for normal cellular signaling, whereas a higher concentration and long-term exposure of ROS cause damage to cellular macromolecules such as DNA, lipids, and proteins, ultimately resulting in necrosis and apoptotic cell death [124]. Altogether, these data suggest that bioenergetic abnormalities are likely to be pathophysiologically relevant to ALS disease.

## 5. Advances in Therapeutic Application of *C. elegans* ALS Models

*C. elegans* has emerged as a valuable platform for investigating potential therapeutic strategies in ALS. Researchers have explored the use of small molecules to modulate disease pathways [125,126,127,128,129,130,131,132,133,134,135], RNA-based therapies targeting specific genes [136,137,138,139,140], and genetic modifiers [81,141] to gain deeper insights into disease mechanisms. In this section, we provide a comprehensive summary of the therapeutic advances in *C. elegans* ALS models, covering various strategies, functions, and mechanisms of treatment. 

As previously discussed, both proteostasis and autophagy play pivotal roles in protein aggregation in ALS, offering a promising avenue for therapeutic intervention. The potential use of medication to uphold proteostasis and boost autophagy holds considerable promise for mitigating the progression of ALS. Trehalose, a disaccharide of glucose found in various organisms, serves as an intriguing example. In *C. elegans*, trehalose treatment resulted in a remarkable extension of the mean lifespan by over 30%, along with enhanced thermotolerance and reduced polyglutamine aggregation [142]. These findings suggest that trehalose exerts its effects by mitigating the aging process and countering internal or external stresses that disrupt proteostasis.

Moreover, trehalose has been demonstrated to protect neurons by inducing autophagy, leading to the clearance of protein aggregates—a concept known as the autophagy induction hypothesis. Several animal studies, including those employing the *C. elegans* model, have shown the activation of autophagy and a reduction in protein aggregates following trehalose administration in neurodegenerative disease models [127]. These collective findings reveal the potential therapeutic benefits of targeting proteostasis and autophagy, with trehalose serving as a compelling candidate for further exploration in the context of ALS treatment. 

These studies not only enhance our comprehension of ALS but also offer insights into potential treatments that may eventually progress to clinical trials for human patients. However, it is imperative to acknowledge that findings in *C. elegans* require validation in more complex models and, ultimately, in clinical trials to ensure their relevance to human ALS. Due to space constraints, we present a summary of the advances in therapeutic approaches in *C. elegans* ALS models in Table 3.

## 6. Limitations of *C. elegans* as ALS Models and Future Directions

ALS is a complex disease influenced by various factors, with an increasing body of literature highlighting the impact of comorbid processes on its pathological progression [5,144,145]. Effectively developing disease-modifying therapies faces a significant challenge due to our limited understanding of the multifaceted pathways contributing to the disease’s development. While mammalian disease models offer valuable in vivo opportunities and share considerable similarities with the human brain, they are accompanied by inherent complexities. On the other hand, the relatively straightforward architecture of *C. elegans*, a microscopic nematode, introduces its own set of constraints. For instance, C. elegans presents limitations for studying systemic pathogenesis of neurodegenerative diseases, particularly in tissues and organ systems crucial for diseases like ALS. Notably, the absence or simplification of structures, such as the central nervous system (CNS) and brain, hinders comprehensive investigations. Moreover, the genetic makeup of *C. elegans* lacks certain features like adaptive immunity and DNA methylation, potentially limiting its representation of these aspects. When employed for biochemical extraction, the use of whole worms can introduce uncertainty regarding tissue-specific signaling. Additionally, the cellular stress responses in nematodes differ from those in mammalian cells. Consequently, *C. elegans* often serves as a complementary model to provide insights into the pathogenesis and therapeutic approaches for ALS. However, it is crucial to emphasize that findings derived from *C. elegans* research require validation in mammalian models and clinical settings to establish their clinical relevance.

One potential strategy for developing *C. elegans* models of ALS entails the functional annotation of human genome variants to discern factors influencing susceptibility and resilience. The burgeoning databases housing human gene sequence information have resulted in an overwhelming volume of variants of uncertain significance. Channeling functional gene analysis in *C. elegans* towards the functional attributes of mutation data can significantly augment our understanding of pathogenic mechanisms and treatment prospects [146]. 

Nematodes possess substantial potential for accelerating the development of neuroprotective drugs due to their straightforward genetic attributes and suitability for high-throughput compound screening [61]. Both target-driven and phenotypic screening approaches can be readily executed in these organisms, making *C. elegans* an exemplary screening target. In contrast to rodent models, worm models provide a swift and cost-effective means to assess numerous drug combinations. Furthermore, the advent of technologies such as CRISPR facilitates the rapid generation of new and more precise nematode models of ALS [147,148]. This is accomplished by precisely delivering a single copy of the identified mutated gene from the patient to the designated location in the worm’s genome. Looking forward, the integration of a more precise genetic *C. elegans* model with a high-throughput automated drug screening platform presents a potentially highly effective strategy for drug discovery in ALS treatment.

In summary, *C. elegans* models have significantly facilitated the transition from experimental research to potential clinical applications. The numerous advantages offered by *C. elegans* present an attractive and ethically sound alternative to more expensive and time-consuming in vitro or mammalian models. The growing track record of translational outcomes resulting from *C. elegans* research positions this microscopic organism to shed light on the remaining uncertainties surrounding ALS. ALS, a pervasive neurodegenerative ailment on a global scale, imposes a substantial burden on tens of millions of individuals daily. Injecting urgency and innovative strategies into model systems research is imperative to expedite discoveries and advancements.

## Figures and Tables

**Figure 1 cells-13-00099-f001:**
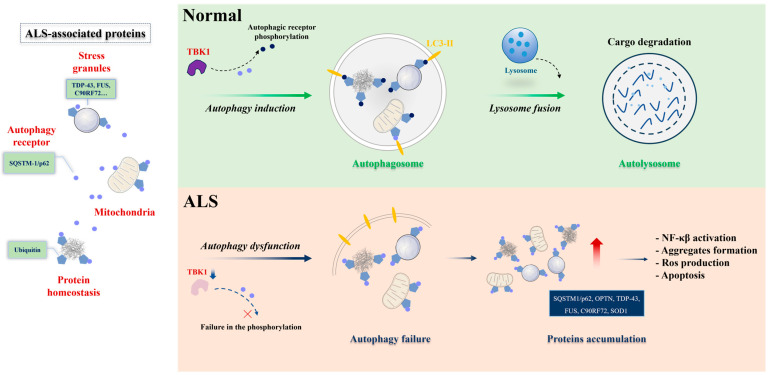
Selective autophagy under physiological and ALS pathological conditions. Protein aggregates, stress granules, and dysfunctional mitochondria serve as substrates for selective autophagy degradation. In physiological conditions (**upper panel**), these substrates are bound by selective autophagy receptors, such as SQSTM-1/p62 (represented in blue circle), via ubiquitin-binding domains (ubiquitin, in pentagon). The selective autophagy receptors associate with LC3-II proteins in the autophagosome (represented in yellow) or other members of the autophagy machinery. Posttranslational modifications in the receptors can enhance binding with ubiquitinated substrates or the LC3-II protein. TBK1 is one of the main kinases acting in this process. The cargo-receptor- LC3-II complexes are then sequestered by de novo double-membrane vesicles called the autophagosome, which fuses with the lysosome for the final degradation. Under ALS conditions (**lower panel**), failure in selective autophagy can occur through mutations in the genes encoding the receptors themselves or in the kinase, reducing the activity of the pathway and promoting the accumulation of toxic substrates for motor neurons. Figure was generated by PowerPoint 2013.

**Table 1 cells-13-00099-t001:** A list of published *C. elegans* models of ALS.

Model	Strain/Transgene Name/Plasmid	Expression in *C. elegans*	Phenotypes
Pro-aggregant lines: *Is*[P*rab-3*::F3ΔK280 + P*myo-2*::mCherry]	BR5270, 5485, 5706, 5944	Constitutive pan-neuronal	Severe locomotive impairment of adulthood at day 1; accelerating aggregate formation; severe developmental deficiency in the nervous system; injury in presynaptic transmission [43]
Anti-aggregant lines: *Is*[P*rab-3*::F3ΔK280(I277P)(I308P) + P*myo-2*::mCherry]	BR5271, 5486, 6427, 6516	Constitutive pan-neuronal	No overt locomotive impairment; minimum influence on neurodevelopment [43]
*Is*[P*hsp-16.2*::s*od-1* (WT, A4V, G37R, G93A) + P*myo-3*::sod-1(WT, A4V)::*gfp* + *rol-6*(su1006)]	n.a.	Heat-shock-inducible muscles	Oxidative stress-induced aggregate formation [44]
*iwIs8*[P*snb-1*::*sod-1*(WT, G85R)::*yfp*]	n.a.	Constitutive pan-neuronal	Severe motor dysfunction is accompanied by both soluble oligomers and insoluble aggregate deposits [45]
*Is*[P*sng-1*::*sod-1*(WT, A4V, G37R, G93C)::*gfp*]	n.a.	Constitutive pan-neuronal	Compared to heterodimers, mutant homodimers demonstrate increased aggregate formation, but G85R heterodimers are more toxic in functional assays [46]
*Is*[P*unc-54*::*sod-1*(WT, G85R, G93A, G127insTGGGstop)::*yfp*]	AM263, 265	Constitutive muscle	SOD-1 mutants demonstrate morphologically heterogeneous aggregates with a variety of biophysical properties and mild motility defects [47]
*ngIs36*[P*unc-25*::*sod-1*(G93A)::*gfp*]	n.a.	GABAergic motor neurons	G93A SOD-1 animals demonstrate progressive motor dysfunction, aggregate formation, and axonal guidance defects [41]
*lin-15(n765ts)*; [P*rgef-1*::FUS (WT, R514G, R521G, R522G, R524S, P525L) + P*pab-1*:: mCherry; *lin-15*(+)]	PJH897	Constitutive pan-neuronal	Forms of cytoplasmic FUS aggregates: R522G, P525L, FUS513, and FUS501 demonstrate a significantly shorter lifespan; P525L, FUS513, and FUS501 demonstrate partially or completely paralyzed, severely shrunken by 8 days of age [48]
*unc-119(ed3)*; *Is*[P*unc-47*::TDP-43-(WT, A315T) + *unc-119*(+)] *unc-119(ed3)*; *Is*[P*unc-47*::FUS-(WT, S57Δ) + *unc-119*(+)]	*xqIs132*, *xqIs133*,	GABAergic motor neurons	Having a normal lifespan, but displayed adult-onset, age-dependent damage to motility, progressive paralysis, neuronal degeneration, and the accumulation of highly insoluble TDP-43 and FUS proteins [49]
*iwIs26*[P*snb-1*::TDP-43-YFP WT], *iwIs22*[P*snb-1*::TDP-C25-YFP], *iwEx20*[P*snb-1*::TDP-43-YFP Q331 K], *iwEx28*[P*snb-1*::TDP-43-YFP M337 V)], *iwIs27*[P*snb-1*::SOD1-YFP WT], *iwIs8*[P*snb-1*::SOD1-YFP G85R]	IW63, IW33, IW20, IW46, IW31, IW8	Constitutive pan-neuronal	Transgenic models have developed robust locomotion defects and protein aggregation [50]

n.a.: no available information.

**Table 2 cells-13-00099-t002:** ALS pathogenic mechanisms.

Pathogenic Molecule	Normal Functions	Pathogenic Mechanism	*C. elegans* Model	Phonotype
C9orf72-SMCR8 complex subunit (C9orf72) [66]	Guanine nucleotide exchange factor (GEF) activity and regulating autophagy [25]	A hexanucleotide repeat (GGGGCC) within the first intron of C9orf72 undergoes expansion with AUG independence, producing five separate dipeptide-containing proteins [37]	*alfa-1* [40]	Motor neuron degeneration and a motility defect [40]
Superoxide dismutase (SOD1) [66]	A cytosolic enzyme, catalyzes the detoxification of superoxide [14]	Mutant alleles of SOD1 generate toxic increases in function in motor neurons; misfold and then eventually aggregate in motor neurons until in vitro; ER stress [67]	a: Pan-neuronal expression of human G85R SOD1 [47]; b: Motor neuron overexpression of a human G93A SOD1 [41]	a: Locomotor deficiency, growth of aggregates and axonal abnormalities [47]; b: Age-dependent paralysis results in the consequence of axonal guidance defects [41]
Transactive response (TAR) DNA-binding protein 43 (TDP-43) [68]	Participate in various steps of RNA metabolism, including mRNA splicing, RNA transport, translation, and microRNA biogenesis [69]	a: Deficiency of normal function in the nucleus; b: A toxic GOF in the form of cytoplasmic aggregates [70]	a: GABAergic neuronal expression of human TDP-43 [49]; b: *C. elegans* homologous gene, TDP-1 [53]	a: Within the GABAergic neurons, there is slowed and uncoordinated movement, as well as degeneration of the motor neurons [49]; b: Deficiency of *tdp-1* results in lower fertility, slower growth, and a locomotor deficit [53]
Progranulin (PGRN)	Participate in a diversity of physiologic and pathological processes that consist of cell proliferation, wound healing, and modulation of inflammation	Decreasing PGRN levels result in the hexanucleotide repeat expansion in the C9orf72 gene	Stress and aging produce PGRN impairing the expression and activity of lysosomal proteases [71]	PGRN deficiency resulted in abnormal expression of multiple lysosomal, immune-related, and lipid metabolic genes lysosomal dysfunction, defects in autophagy, and neuroinflammation [72]
RNA-binding protein FUS/TLS (FUS)	DNA repair and several aspects of RNA metabolism involving transcription, alternative splicing, mRNA transport, mRNA stability, and microRNA biogenesis [73]	Disturb the nuclear localization signal, resulting in the mislocalization of FUS to the cytoplasm with protein aggregates [74]	a: Expressing a FUS variant prone to aggregate in GABAergic neurons by the *unc-47* promoter [49]; b: Expressing panneuronlly in FUS mutants under control of the rgef-1 promoter [48]; c: *C. elegans* homologous gene, *fust-1* [48,75]	a: Neurodegeneration, synaptic dysfunction, paralysis and aggregation; b: Motor dysfunction; c: Achieve maximum microRNA (miRNA)-mediated gene silencing [76]
NIMA-related serine/threonine kinase protein family (NEK), NEK1 [68]	Controlling the cell cycle, DNA damage repair, ciliogenesis splicing, RNA transport, translation, and microRNA biogenesis [69,77]	Increasing DNA damage and a compromised DNA damage response [78]	Acting on DDR signaling downstream of ATM/ATR [77]	DNA damage response and repair as well as mitochondrial function [78]

**Table 3 cells-13-00099-t003:** Advances in Therapeutic Strategies.

Therapeutic Strategies	References	Functions	Mechanisms of Treatment
Small molecules	Riluzole [125,126]	Decreasing glutamate release for neuroprotective	Decreasing glutamate release for neuroprotective
	Trehalose [127]	Autophagy-enhancing properties contribute to clear protein aggregates in ALS	Improving motor function and increasing the lifespan of *C. elegans* models of ALS
	Curcumin [128]	Decreasing oxidative stress and slowing disease progression	Prospective neuroprotective effects
	Methylene blue [129,130]	An aggregation inhibitor of the phenothiazine class	Protects against oxidative stress
	Bafilomycin [131]	Blocking autophagosome-lysosome fusion and inhibiting acidification and protein degradation in cell lysosomes to produce the effect of inducing apoptosis	Decreasing neurodegeneration via inhibiting autophagic vesicle maturation
	Dantrolene [132,133]	A muscle relaxant for noncompetitively inhibiting human erythrocyte glutathione reductase	Decreasing neurodegeneration by inhibiting intracellular calcium release in the ER
	Probucol [134]	Regulating blood lipids and anti-lipid peroxidation	Attenuating neurodegeneration with its antioxidant properties
	Resveratrol [135,143]	Antioxidant and anti-inflammatory properties	Mitigating ALS-like symptoms via activating cellular protective mechanisms
RNA-based therapies	RNAi (RNA Interference) [136,137,138,139]	Gene therapy for ALS and FTD is possible because of the reduction in toxicity induced by the repeat-containing C9orf72 transcripts	Aiming and knocking down genes associated with ALS-related proteins by RNAi
	Antisense Oligonucleotides (ASOs) [139,140]	Reducing, restoring, or modifying RNA and protein expression	Modulating the expression of ALS-associated genes and potentially reducing toxic protein production

## Data Availability

No new data were created in this study.

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
