# Peer review of "Amyotrophic Lateral Sclerosis Mechanism: Insights from the Caenorhabditis elegans Models"

_cells, 2024, doi:10.3390/cells13010099_

Round 1

Reviewer 1 Report

Comments and Suggestions for Authors

In the present review article, the authors compiled C. elegans-based ALS disease models using the various genetic causative agents associated with ALS patients. This review has done a comprehensive analysis on C. elegans transgenic models with tissue-specific types or neuronal subtypes in which the disease factors are expressed and the linked phenotypes of the disease model. This study provides nice information on the mechanisms associated with the ALS disease models and the advances in the ALS therapeutics identified by using C. elegans ALS-based disease models.

 Major points:

Section 3:

1.        The paragraph that appears between lines 115 and 123 would benefit from revision for clarity. I think, if I read correctly, that a phenotype occurs when there is a tissue-specific expression, and that a phenotype results from combining YFP with SOD-1, subject to whether YFP actually caused the phenotype? But it was hard to say.

2.        It is recommended to modify the title. For example-It could be amend as follows: The genetic factors and the associated disease mechanisms in the C. elegans disease models (as this section primarily covers the genetic factors description and associated  gene mutations, addiontionally, it addresses what phenotypic events occur when these are expressed in the C. elegans). However, I feel that a cellular mechanism is addressed in the following section (Cell signaling ptahways/section4); it is not clear how authors have integrated the “mechanism investigation” in this title.

3.        In Line 134, I am unable to understand how C. elegans TDP-1 played a “pioneering role” for human TDP-43 in C. elegans. This seems like it might be over-claiming.

4.        In line 154, the authors are discussing FUS and then switch to dynactin. Context is needed as to what these two genes have to do with each other (if anything).

5.        In line 163, the authors discuss the characteristics/comparison of the C9ORF72, and its C. elegans homolog alfa-1 (noting the lack of expanding nucleotide repeats in alfa-1). It could be useful to have similar comparative analyses for additional geneic factors like FUS and TDP. Are their protein domain structures completely conserved from worm to human?

6.        In line 178, ref. 50 , this reference is unclear to me and doesn't appear to have anything to do with C. elegans. I think that the sentence needs to be reframed or the citation fixed. In general a number of citations seem to have been mixed up / out of order?

Section 4:

7.        The cell signaling underlying ALS-based diseases are addressed in detail in Section 4, with a primary emphasis on two aspects: immune regulation and autophagy. It is unknown, though, if these are the only two major processes for ALS-based disorders, or whether there are additional signaling mechanisms as well. For example, neurons expressing mutant TDP-43/FUS variant shows altered neuronal physiology (energy metabolism/ neurite growth) etc., if included in a paragraph, it will improve the scope of manuscript. Perhaps this can be discussed, but, with the caveat that they are outside the purview of this section because of issues like limited space?

8.        Manuscript Title: I don’t think the second clause of the title reads well. In fact I don’t understand the meaning. For better impact of the article perhaps instead of “… the C. elegans Models Count” it could be something like “… insights from C. elegans models”

Comments on the Quality of English Language

Most important is that the title does not read well (see attached notes). Other than that, a few sentences are difficult to parse but overall it's okay.

Reviewer 2 Report

Comments and Suggestions for Authors

In this review, Chen et al. discussed the current state of modeling ALS in C. elegans and the proteins and molecules associated with ALS. Although the effort to systematically review the field is highly appreciated, I have multiple concerns about the review that limit my enthusiasm to publish it in its current form.

Major concerns:

1.  The quality of the writing needs to be drastically improved. The text is often incorrect and confusing due to language problems. I highly recommend the authors to find professional editing services to correct language problems. I tried my best to list some below but there are just way too many.

2.   Sections 2 and 3 should be reorganized. One can be for “ALS-associated proteins” and the other for “C. elegans models of ALS”. In the current setup, section 2 is too short and only has some general information without specifics. Section 3 includes both the list of ALS-associated proteins (most information is from clinical studies or mammalian models) and how they are used to model ALS in C. elegans. Separating the two will improve the readability of the paper.

3.     Section 4 should be renamed as “Pathogenic mechanisms of ALS” and three subheadings can be used for the three aspects the authors discussed, namely 1) innate immunity; 2) autophagy; and 3) protein homeostasis. However, none of the three aspects are discussed with enough detail to provide the readers with important insights. Also, the role of mitochondrial dysfunction in ALS can be discussed (although there is a recent review on the topic in this journal). I recommend the authors to draw inspiration from the other review to improve the scientific content, writing style, and illustrations of the current manuscript.

https://www.ncbi.nlm.nih.gov/pmc/articles/PMC9265651/)

Line 193-194: It is important to note that although C. elegans has an innate immune system, it does not have a classical inflammatory response or inflammatory cytokines like in mammals. So, it will be helpful if the authors can discuss more about the mechanisms by which innate immunity in C. elegans models modulates neurodegeneration.

Line 210-231: The role of autophagy in ALS is not clear from this paragraph. It started by saying that there are increased autophagy markers and disrupted transport of autophagosome (leading to accumulation) in ALS motor neurons, suggesting that there may be elevated autophagy in ALS. Then it switches to state that defects in autophagy contribute to neurodegeneration, but the defect is shown as the accumulation of autophagy adaptor protein SQSTM1/SQST-1. I don’t get the logic. Does the SQSTM1 inhibit autophagy? If some autophagy genes negatively regulate autophagy, I think the authors need to have a Figure to illustrate the regulation of autophagy and at which step all the mentioned proteins work. Overall, it is not very clear to me whether increased or decreased autophagy causes neurodegeneration in ALS and what the mechanism is behind the phenomenon.

Line 232-253: The discussion about protein homeostasis is inadequate. It mentioned HSF-1 and Insulin signaling (DAF-2) but it never discusses how these signaling pathways regulate protein degradation or homeostasis. Do the signaling pathways work through protein ubiquitination or proteosome-mediated degradation? Any specific targets are involved?

4.     The discussion for the development of therapeutics for ALS using the C. elegans model is inadequate. Section 5 from Line 254-263 is too short. No specific therapeutic approach is mentioned in the text. The authors need to at least describe the content of Table 3 and choose one or two from each category to discuss in great depth. Also, “genetic modifiers” should not be considered as therapeutic approaches.

Comments on the Quality of English Language

Minor concerns on language (a few examples below) and the need for clarifications.

Line 23: Missing a period.

Line 28-31: This sentence has some problems; The phrase “insights from …” is not connected with the preceding part of the sentence.

Line 56: “proteins”?? What proteins? Do you mean “disease-associated proteins”? This needs to be specified.

Line 60: “mutants” of what?? “ALS-related genes”?

Line 67-68: incorrect or at least very awkward expression.

Line 77: “Frequently encountered proteins”?? “encountered by what?”

Line 86: “subsequent paralysis”?? I think “subsequent neuronal death” is better.

Line 103: The sentence does not make sense. The beginning “C9ORF72,” needs to be deleted.

Line 105: “AUG individual translation” ?? Do you mean “AUG-independent translation”?

Line 117: “to introduce” may be changed  “to express”.

Line 123, did the muscle expression of SOD1::YFP cause any phenotype?

Line 130, “Expressing SOD-1loss of function mutants”?? Did the author mean Loss-of-function mutations in sod-1 or overexpression of SOD-1 loss-of-function mutants in the wild-type background that contains a functional sod-1 copy?

Line 143, “the overactivation of wild-type TDP-43”? What is overactivation? Do you mean overexpression?

Line 146: “serves as” should be changed to “is”.

Line 154-157: Was the dnc-1 knockdown conducted on top of the expression of mutant FUS or independent of it? If later, how is dnc-1 downregulation related to ALS?

Line 165-167: Did the alfa-1 mutants show earlier onset or more severe phenotype of aging-related motor impairment than the wild-type?

Line 174: Can the authors comment on the potential reason for weaker phenotype with longer GGGGCC repeats? This appears to be counterintuitive.

Line 175: “Gene mutations encoding optineurin …” should be changed to “Mutations in the OPTN gene encoding optineurin …”

Line 178: These mutations were found in patients or C. elegans? Are they just disease-associated mutations that were modeled in C. elegans?

Line 178-179: “reversal of the inhibition of NF-kB activation” may be changed to “activation of NF-kB”.

Line 180: “These mutations demonstrated that” should be deleted.

Line 181: How distinct the cytoplasmic distribution is from the wild type? I think this is an important piece of information to disclose.

Line 213-217: Dynactin links motors to all sorts of organelles and cargos. Knocking down dynactin can affect the transport of not only autophagosomes but also many other organelles. So, is there any evidence to support the effects to be caused by the accumulation of autophagosomes in the cell body?? By the way, the author mentioned “increased number of autophagosomes”. Does it mean when autophagosomes cannot be transported, they somehow increase in numbers? The logic is not very clear to me.

Line 224: “ALS FUS animals” means the human FUS mutant-overexpressing animals? Say it, if this is the case.

Line 227: Does it mean SQST-1 works through an autophagy-independent pathway or else? If this point is important, the authors should add more explanation.

Line 246-247: It is better to say, “the control of insulin/IGF-1 signaling pathway by phosphorylation”.

Reviewer 3 Report

Comments and Suggestions for Authors

The manuscript “Amyotrophic Lateral Sclerosis Mechanism, the Caenorhabditis elegans models count” by Chen et al. is a comprehensive review of our current understanding of using C. elegans as a model to study amyotrophic lateral sclerosis (ALS). The authors discussed what we know about the molecular basis of ALS from human (patient) samples and how a majority of that can be recapitulated in different C. elegans models of ALS. The authors briefly described the characteristics and phenotypes of these models and different signaling pathways that are perturbed in these models. Authors also discussed the limitations of using C. elegans as a model to study neurodegeneration and the future perspectives. Overall, I thoroughly enjoyed reading the manuscript and appreciate the authors’ effort. I have some minor concerns listed below that should be addressed.

Specific points:

1.      Section 3 should ideally be divided into two parts – (a) molecular basis of ALS and (b) molecular mechanisms of ALS observed in C. elegans.

2.      A figure summarizing cellular and molecular mechanism underlying ALS will be helpful.

3.      The second paragraph of section 2 (ALS associated genetic mutations modeled in C. elegans) describing why C. elegans is a powerful tool to investigate the mechanism underlying neurodegenerative diseases should come first, followed by the first paragraph describing methods of making C. elegans model of ALS.

4.      There are some grammatical and typographical errors that need to be corrected.

Comments on the Quality of English Language

N/A.

Round 2

Reviewer 2 Report

Comments and Suggestions for Authors

Comments on the revised manuscript:

I appreciate the authors’ response to my concerns and the efforts they put into improving the manuscript. It is a much better paper. I have a few minor suggestions on the text before it can be accepted.

1.     Line 127, “C. elegans” needs to be italicized. I recommend the authors go through the text carefully one more time since there are still some typos and errors.

2.     Line 136, It is better to move “under the control of synaptobrevin (snb-1) gene promoter” to the place after “pan-neuronal expression of SOD-1(G85R)” on line 134.

3.     Line 161, “neuronal degradation” should be changed to “neuronal degeneration”.

4.     Line 196-206, from the text, it is not clear how mutations in OPTN are modeled in C. elegans since the author did not mention either the C. elegans homolog of OPTN or any transgenic models in C. elegans.

5.     Line 222-224, this sentence needs to be rephrased because the logic is incorrect here. Activation of immune response genes does not necessarily indicate neurodegeneration; it could be caused by pathogenic infection. So, it is better to say that the expression of TDP-43 or FUS triggers the upregulation of immune response genes, suggesting that innate immune response may contribute to motor neuron neurodegeneration.

6.     Line 249, Do the cited studies indicate whether the increased LC3 staining occurs in the cell body or axon or both of the spinal motor neurons? The same goes for Lines 250-252, when dnc-1 is knocked down, does the accumulation of autophagosomes accumulate in the cell body or axon or both? This information is important since it can tell where autophagosomes normally function and the dysregulation of its spatial distribution may be an underlying pathogenic mechanism of ALS.

7.     Line 262, I think this can be explained a bit more. The authors can say that active turnover of SQSTM1/p62 typically signifies functional autophagy and balanced proteostasis, whereas abnormal accumulation of SQSTM1/p62 indicates dysfunctional autophagy. Cite Figure 1 and the necessary references to support this point.

8.     Lines 270-274, since sqst-1 is an important autophagy receptor, deleting it is not likely to restore autophagy. But this result somehow indicates SQST-1 functions independently of autophagy, which directly challenges the model the authors are presenting in Figure 1. I think some potential alternative hypotheses need to be discussed here to let the readers understand the role of SQST-1 in ALS better.

9. In Figure 1, the authors indicated that some kinase is downregulated in ALS, which may cause the failure in packaging the protein aggregates and damaged organelles into the autophagosome, but this point is not mentioned in the main text at all. If this is a major mechanism for the failure in autophagy in ALS, the author should discuss the evidence more.

10.  Line 315, delete “is” or “serves”.

11.  Line 326, about the “Energy metabolism” section. The author appears to present two seemingly conflicting mechanisms and does not reconcile them with proper discussions. AMPK promotes catabolic metabolism and ATP production, which somehow contribute to ALS, whereas mitochondrial dysfunction causes energy failure and defects in ATP production in the cells and also contributes to ALS. The authors need to at least acknowledge the conflicting theories here and try to come up with some potential explanations.

Comments on the Quality of English Language

The language has improved a lot in the revised manuscript, although there are still some typos and errors that need to be corrected. 
